# Prognostic Impact of Serum Free Light Chain Ratio Normalization in Patients with Multiple Myeloma Treated within the GMMG-MM5 Trial

**DOI:** 10.3390/cancers13194856

**Published:** 2021-09-28

**Authors:** Eva-Maria Klein, Diana Tichy, Hans J. Salwender, Elias K. Mai, Jan Duerig, Katja C. Weisel, Axel Benner, Uta Bertsch, Mabast Akhavanpoor, Britta Besemer, Markus Munder, Hans-Walter Lindemann, Dirk Hose, Anja Seckinger, Steffen Luntz, Anna Jauch, Ahmet Elmaagacli, Stephan Fuhrmann, Peter Brossart, Martin Goerner, Helga Bernhard, Marc S. Raab, Igor W. Blau, Mathias Haenel, Christof Scheid, Hartmut Goldschmidt

**Affiliations:** 1Department of Medicine V, Hematology, Oncology and Rheumatology, University of Heidelberg, 69120 Heidelberg, Germany; elias.mai@med.uni-heidelberg.de (E.K.M.); uta.bertsch@med.uni-heidelberg.de (U.B.); m-moohialdin@gmx.de (M.A.); dirk_hose@yahoo.de (D.H.); anja.seckinger@gmx.de (A.S.); marc.raab@med.uni-heidelberg.de (M.S.R.); hartmut.goldschmidt@med.uni-heidelberg.de (H.G.); 2Department of Internal Medicine 5, Klinikum Nuremberg, Paracelsus Medical University, 90419 Nuremberg, Germany; 3Division of Biostatistics, German Cancer Research Center (DKFZ), 69120 Heidelberg, Germany; d.tichy@dkfz-heidelberg.de (D.T.); benner@dkfz-heidelberg.de (A.B.); 4Asklepios Tumorzentrum Hamburg, AK Altona and AK St. Georg, 22763 Hamburg, Germany; h.salwender@asklepios.com; 5Department of Hematology, University Clinic Essen, 45147 Essen, Germany; jan.duerig@uk-essen.de; 6Department of Oncology, Hematology and Bone Marrow Transplantation with Section of Pneumology, University Medical Center Hamburg-Eppendorf, 20246 Hamburg, Germany; k.weisel@uke.de; 7National Center for Tumor Diseases, 69120 Heidelberg, Germany; 8Department of Hematology, Oncology and Immunology, University Hospital Tübingen, 72076 Tübingen, Germany; britta.besemer@med.uni-tuebingen.de; 9Department of Internal Medicine III, University Medical Center Mainz, 55131 Mainz, Germany; munder@uni-mainz.de; 10Department of Hematology and Oncology, Katholisches Krankenhaus Hagen, 58097 Hagen, Germany; w.lindemann@kkh-hagen.de; 11Coordination Centre for Clinical Trials (KKS) Heidelberg, 69120 Heidelberg, Germany; steffen.luntz@med.uni-heidelberg.de; 12Institute of Human Genetics, University of Heidelberg, 69120 Heidelberg, Germany; anna.jauch@med.uni-heidelberg.de; 13Department of Hematology and Oncology, Asklepios Hospital Hamburg St. Georg, 20099 Hamburg, Germany; a.elmaagacli@asklepios.com; 14Department of Hematology and Oncology, Helios Hospital Berlin Buch, 13125 Berlin, Germany; stephan.fuhrmann@helios-gesundheit.de; 15Department of Internal Medicine, Oncology, Hematology, Immuno-Oncology and Rheumatology/Clinical Immunology, University Hospital Bonn, 53127 Bonn, Germany; peter.brossart@ukb.uni-bonn.de; 16Department of Hematology, Oncology and Palliative Care, Klinikum Bielefeld, 33604 Bielefeld, Germany; martin.goerner@klinikumbielefeld.de; 17Internal Medicine V, Klinikum Darmstadt, 64283 Darmstadt, Germany; helga.bernhard@mail.klinikum-darmstadt.de; 18Medical Clinic, Charité University Medicine Berlin, 13353 Berlin, Germany; igor.blau@charite.de; 19Department of Internal Medicine III, Klinikum Chemnitz, 09116 Chemnitz, Germany; m.haenel@skc.de; 20Department of Internal Medicine I, University Hospital Cologne, 50937 Cologne, Germany; c.scheid@uni-koeln.de

**Keywords:** multiple myeloma, prognostic factors, serum free light chain ratio normalization, immune reconstitution, time-dependent analysis

## Abstract

**Simple Summary:**

For multiple myeloma (MM) patients with measurable disease, there is no recommendation to monitor serum free light chains during therapy. However, this could provide important information in terms of prognosis. We investigated the prognostic impact of serum free light chain ratio (FLCr) normalization in 590 patients with secretory MM during first-line treatment within the German-Speaking Myeloma Multicenter Group MM5 trial. We are able to show that there is an increasing percentage of patients who achieve FLCr normalization during therapy. Importantly, we demonstrate that FLCr normalization at any time before the start of maintenance is significantly associated with prolonged progression-free and overall survival in multivariable time-dependent Cox regression analyses. This suggests that FLCr normalization during therapy is an important and simple way to assess prognostic factor in MM and supports the serial measurement of serum free light chains during therapy, even in patients with secretory MM.

**Abstract:**

We investigated the prognostic impact of time-dependent serum free light chain ratio (FLCr) normalization in 590 patients with secretory multiple myeloma (MM) during first-line treatment within the German-Speaking Myeloma Multicenter Group MM5 trial. Serum free light chains (sFLC) were assessed by the Freelite test at baseline, after induction, mobilization, autologous blood stem cell transplantation, consolidation and every three months during maintenance or follow up within two years after the start of maintenance. The proportion of patients with a normal or normalized FLCr increased from 3.6% at baseline to 23.2% after induction and 64.7% after consolidation. The achievement of FLCr normalization at any one time before the start of maintenance was associated with significantly prolonged progression-free survival (PFS) (*p* < 0.01, hazard ratio (HR) = 0.61, 95% confidence interval (95% CI) = 0.47–0.79) and overall survival (OS) (*p* = 0.02, HR = 0.67, 95% CI = 0.48–0.93) in multivariable time-dependent Cox regression analyses. Furthermore, reaching immune reconstitution, defined as the normalization of uninvolved immunoglobulins, before maintenance was associated with superior PFS (*p* = 0.04, HR = 0.77, 95% CI = 0.60–0.99) and OS (*p* = 0.01, HR = 0.59, 95% CI = 0.41–0.86). We conclude that FLCr normalization during therapy is an important favorable prognostic factor in MM. Therefore, we recommend serial measurements of sFLC during therapy until achieving FLCr normalization, even in patients with secretory MM.

## 1. Introduction

Multiple myeloma (MM) is a cancer of the bone marrow, characterized by a clonal proliferation of plasma cells producing monoclonal protein. The monoclonal protein can be either a complete immunoglobulin consisting of two heavy and two light chains or, in the case of light chain MM, light chains only. However, in patients with lgG, lgA, lgM, lgD or lgE MM, more light chains are also produced than heavy chains, leading to a measurable increase in light chains in the serum and also in the urine after the renal reabsorption capacity is exceeded [1]. 

An abnormal serum free light chain ratio (kappa/lambda, FLCr) can be found in approximately 95–98% of patients with newly diagnosed MM [2,3]. The prognostic impact of the FLCr at diagnosis was demonstrated in several clinical trials [2,3,4,5]. Therefore, according to the recommendations of the International Myeloma Working Group (IMWG), serum free light chains (sFLC) should be assessed at baseline in MM [6]. Further indications for measuring sFLC in MM are the screening for MM, the monitoring of a part of the patients with previously called asecretory MM (involved sFLC ≥ 100 mg/L provided an abnormal FLCr) and the determination of the stringent complete response (sCR) [6,7,8]. For MM patients with measurable disease by a serum monoclonal protein ≥10 g/L or a light chain excretion in the 24 h urine ≥200 mg per day, there is no recommendation for monitoring of sFLC during therapy yet [9]. 

Nonetheless, sequential measuring sFLC in these patients might be beneficial. Lopez-Anglada and colleagues were able to demonstrate in an analysis on patients treated in three phase-three trials of the PETHEMA/GEM that achieving a normalization of the FLCr after treatment is associated with prolonged progression-free survival (PFS) and overall survival (OS) and that the persistence of an involved sFLC ≥100 mg/L after the end of treatment has a negative prognostic impact [10]. Another analysis showed that in MM patients achieving no complete response (CR) during first-line therapy, an FLCr normalization at the time point of best response also has an independent beneficial effect on PFS and OS [11]. Furthermore, Dejoie and colleagues recently published a proposal for a modification of the IMWG response criteria, replacing 24 h urine measurements with sFLC in the response assessment of patients with secretory MM [12].

Besides an abnormal FLCr, immunoparesis is a hallmark of MM since the expansion of clonal plasma cells leads to a displacement of normal plasma cells and therefore decreased production of polyclonal immunoglobulins. Accordingly, immunoparesis accounts for an adverse prognosis in newly diagnosed, as well as relapsed, MM [13,14,15]. In turn, achieving a reconstitution of polyclonal immunoglobulins during the course of therapy is associated with prolonged PFS and OS. The recovery of polyclonal immunoglobulins one year after autologous blood stem cell transplantation (ASCT) is associated with a favorable prognosis [16].

The aim of our study is to investigate FLCr normalization during the course of therapy within the German-Speaking Myeloma Multicenter Group (GMMG) MM5 trial [17] and its impact on PFS and OS. In addition, we evaluate the importance of achieving immune reconstitution during therapy. 

## 2. Materials and Methods

### 2.1. MM5 Trial

Newly diagnosed transplant-eligible MM patients with measurable disease were included in the prospective multicenter phase-three trial MM5 (EudraCT No. 2010-019173-16) and randomized to one of four different treatment arms. Eligibility criteria, design and primary endpoints of the trial have been published [17,18]. After randomization, patients received induction therapy with three cycles of bortezomib, doxorubicin and dexamethasone (PAd, arms A1 and B1) or bortezomib, cyclophosphamide and dexamethasone (VCD, arms A2 and B2). Then stem cell mobilization and subsequent melphalan high-dose therapy and ASCT were performed according to local protocols. Afterwards, lenalidomide consolidation and maintenance were conducted. Patients received lenalidomide maintenance for two years in arms A1 and A2 or until the achievement of CR in arms B1 and B2 (Appendix A; Appendix A). The MM5 trial was approved by the local ethics committees of all participating centers (leading ethics committee University of Heidelberg AFmu-119/2010). All patients gave written informed consent.

### 2.2. Assessment of sFLC and Immunoglobulins

The Freelite test (The Binding Site Group Ltd., Birmingham, Great Britain) was used to prospectively quantify sFLC centrally at inclusion, after induction, stem cell mobilization, ASCT, consolidation and every three months during maintenance or follow up within two years after the start of maintenance [19]. The immunoglobulins IgG, IgA and IgM were prospectively assessed at the same time points. For the definition of FLCr normalization, the established reference range by Katzmann et al. for the kappa/lambda ratio of 0.26–1.65 was used [20]. In the case of renal insufficiency (creatinine >2 mg/dL and/or glomerular filtration rate <40 mL/min), the adapted range 0.37–3.1 for FLCr was applied [21]. Immunoparesis was defined by the suppression of at least one uninvolved immunoglobulin [22]. For the achievement of immune reconstitution, the normalization of all uninvolved immunoglobulins was required. The following reference ranges were used: IgG 7.0–16.0 g/L, IgA 0.7–4.0 g/L and IgM 0.4–2.3 g/L.

### 2.3. Statistical Methods

The achievement of FLCr normalization was determined for patients on study at baseline, after induction, stem cell mobilization, ASCT, consolidation and every three months during maintenance or follow up until the end of the study. Thus, the values of FLCr normalization are yes, no and missing. In the second step, we consolidated the received information and determined if FLCr normalization was achieved at any time until the start of maintenance at the latest, irrespective of whether the achievement was lost in between. Thereby, the date of first achievement was used to model the time from randomization to the first achievement of FLCr normalization. The achievement of immune reconstitution was analogously determined. In addition, the achievement of CR after consolidation was assessed.

A multivariable Cox regression model with time-dependent covariates was applied to analyze the impact of FLCr normalization and immune reconstitution until the start of maintenance at the latest on PFS and OS. FLCr normalization, immune reconstitution and CR after consolidation were modeled as time-dependent covariates. The set of fixed covariates consisted of age, the International Staging System (ISS), cytogenetic risk and treatment arm. The Simon–Makuch estimators were derived to present the estimated risk of progression or death under the state of achieved FLCr normalization or immune reconstitution [23]. The Simon–Makuch plots show the impact of time-dependent variables on PFS and OS, taking the time-dependent change of the variables into account. Here, the survival times until a potential achievement of FLCr normalization/immune reconstitution and after a potential achievement are separately shown in two curves. Therefore, it is possible that one patient can be found in two curves. Patients who never achieve FLCr normalization are shown in the curve “before FLCr normalization” as well as patients who start with an abnormal FLCr and achieve a normalization during therapy. Then the patients who achieve a normalization can be found in the curve “after FLCr normalization” as well as patients who already started with a normal FLCr.

Furthermore, the prognostic impact of FLCr normalization at end of induction and consolidation, respectively, and immune reconstitution after consolidation were assessed by an equivalent multivariable Cox regression model.

To evaluate a previously described prognostic effect of FLCr at diagnosis, the impact of an FLCr of 1/32-32 vs. <1/32 or >32 at baseline on PFS and OS was examined in a univariate Cox regression analysis, according to Snozek et al. [2]. The curves for PFS and OS and the corresponding 95% confidence interval (CI) were derived using the Kaplan–Meier method [24].

Values of *p* <0.05 were considered statistically significant. The analyses were conducted using R version 3.6.2 (https://www.R-project.org, accessed on 17 September 2021). 

## 3. Results

### 3.1. Patient Cohort

The expanded population of the MM5 trial consisted of 604 patients [25]. Three of them were excluded due to a violation of the inclusion criteria. Among 601 patients of the ITT population, a number of 590 patients were evaluable for multivariable time-to-event analysis and made up the corresponding analysis population. For 11 patients, FLCr normalization or immune reconstitution could not be determined due to missing values of sFLC, immunoglobulins, creatinine, glomerular filtration rate or missing an assessment date of sFLC or immunoglobulins. 

The baseline characteristics of the analysis population (*n* = 590) can be found in Table 1. After the end of consolidation, 126 of 456 patients on study at the start of the second cycle of consolidation achieved a CR and 320 a non-CR. Ten patients on study had no response assessment after consolidation.

An FLCr between 1/32 and 32 vs. <1/32 or >32 at baseline was associated with prolonged PFS (*p* = 0.01, hazard ratio (HR) = 0.74, 95% CI = 0.59–0.94) and OS (*p* = 0.01, HR = 0.62, 95% CI = 0.44–0.88) in univariate Cox regression analyses (Figure 1).

### 3.2. FLCr Normalization during Therapy in the MM5 Trial and Its Impact on PFS and OS

The percentage of patients with a normal FLCr increased from 3.6% (21/590) at baseline to 23.2% (131/564) after induction, 48.5% (249/513) after ASCT and 64.7% (295/456) after consolidation therapy (Figure 2). During maintenance therapy, the percentage slowly decreased from 61.7% (263/426) after three months to 55.4% (209/377) after 12 months to 48.1% (140/291) after 24 months.

Among 590 evaluable patients for multivariable regression analyses, 401 patients achieved a FLCr normalization at any time point before the start of maintenance. A normalization of the FLCr until the start of maintenance at the latest significantly prolonged PFS (*p* < 0.01, HR = 0.61, 95% CI = 0.47–0.79) and OS (*p* = 0.02, HR = 0.67, 95% CI = 0.48–0.93) in the multivariable time-dependent Cox regression analyses (Table 2). This impact was not associated with a deep response (CR vs. non-CR) after consolidation. Furthermore, ISS II and III compared to ISS I and the presence of high-risk cytogenetics were significantly associated with an inferior PFS and OS. Lenalidomide maintenance until the achievement of CR (study arms B1 and B2) compared to a fixed duration of two years (study arms A1 and A2) was linked to a shorter OS (Table 2). Figure 3A,B present the Simon–Makuch estimators on the risk of progression and death depending on the achievement of FLCr normalization. 

Next, we assessed the impact of a FLCr normalization at the predefined time points “after induction” and “after consolidation” on PFS and OS. Achieving a FLCr normalization after induction showed no influence on PFS (*p* = 0.11, HR = 0.81, 95% CI = 0.62–1.05) and OS (*p* = 0.16, HR = 0.75, 95% CI = 0.50–1.12) in multivariable analyses (Appendix A). Similar results were seen at the time point after consolidation (PFS: *p* = 0.12, HR = 0.82, 95% CI = 0.64–1.05, OS: *p* = 0.34, HR = 0.85, 95% CI = 0.60–1.19) (Appendix A). 

### 3.3. Achievement of Immune Reconstitution during Therapy in the MM5 Trial and Its Impact on PFS and OS

At baseline, 9.0% (53/590) of the patients had normal immunoglobulins, and the amount further decreased to 2.0% (11/564) after induction due to therapy. Immune reconstitution was noted in 15.4% (79/513) of patients after ASCT and in 32.2% (147/456) after consolidation therapy (Figure 4).

In total, 227 of 590 patients evaluable for multivariable analyses achieved an immune reconstitution at any time point before the start of maintenance. The achievement of immune reconstitution until the start of maintenance at the latest significantly prolonged PFS (*p* = 0.04, HR = 0.77, 95% CI = 0.60–0.99) and OS (*p* = 0.01, HR = 0.59, 95% CI = 0.41–0.86) in the multivariable time-dependent Cox regression analyses (Table 3). Figure 3C,D show the Simon–Makuch estimators regarding the risk of progression and death depending on immune reconstitution until maintenance.

The achievement of immune reconstitution after consolidation was associated with a significantly prolonged OS (*p* < 0.01, HR = 0.54, 95% CI = 0.36–0.83) (Appendix A). The effect on PFS was not significant (*p* = 0.15, HR = 0.83, 95% CI = 0.64–1.07) (Appendix A). 

## 4. Discussion

Serial measurements of sFLC during therapy in patients with secretory MM have not been recommended by the IMWG so far [6]. However, these measurements might provide important information in terms of prognosis. The sFLC represent tumor burden and have a shorter half-life (T1/2) than immunoglobulins (T1/2 sFLC 2–6 h, T1/2 IgG 20–25 d), allowing an earlier evaluation of the response to therapy [26,27,28]. Furthermore, due to intraclonal heterogeneity in MM, two to ten percent of the patients develop a sFLC escape at relapse [29,30,31]. Finally, the measurement of light chain excretion in 24-hour urine samples for a response assessment in MM remains controversial because of its dependence on renal function and correct urine collection in the clinical routine [12].

In the present study, we analyzed the prognostic impact of a time-dependent FLCr normalization during the course of therapy in newly diagnosed patients with secretory MM treated within the GMMG MM5 trial. To our knowledge, this is the first study assessing the impact of time-dependent FLCr normalization during therapy in a large cohort of patients with secretory MM.

As expected, our analysis demonstrates that during the course of first-line therapy, there is an increasing percentage of patients achieving FLCr normalization with a maximum after consolidation. Tacchetti et al. showed similar rates of patients achieving a normal FLCr after first-line treatment with a bortezomib-based regime [32]. In contrast, in a Japanese study, only 41% of the patients reached FLCr normalization after a novel agent-containing treatment. However, this may be explained by the small number of patients receiving ASCT (<20%) and that there were only a few available novel agents in the time period 2004–2012 [33].

Furthermore, we show, for the first time, that achieving a time-dependent FLCr normalization at any time point before the start of maintenance has a strong beneficial effect on PFS and OS, independent of age, ISS, cytogenetics, treatment arm and even a deep response after consolidation. In contrast, a FLCr normalization at the defined time points after induction and after consolidation alone has no prognostic significance. A possible explanation for this might be that the prognostic effect of FLCr normalization is time-dependent and cannot be attributed to a defined time point during therapy. In addition, by defining certain time points, the event of reaching FLCr normalization at one time point might be too low to reach statistical significance. In contrast, Lopez-Anglada et al. demonstrated that a normal FLCr after induction or after ASCT is associated with a prolonged PFS and OS [10]. The differences might be explained by the number of patients and different variables in the multivariate models. Based on our analyses, we propose serial measurements of sFLC during MM therapy, at least until the achievement of FLCr normalization but preferably also after FLCr normalization to detect a relapse with sFLC escape. 

Other studies assessing time-dependent FLCr normalization in secretory MM are not available in the current literature, but there are studies similarly demonstrating a significant effect of a FLCr normalization after therapy on PFS and OS [10,33,34]. Furthermore, Alhaj Moustafa et al. were able to demonstrate a positive prognostic impact of FLCr normalization independent of the response in patients with secretory MM who do not achieve CR in the first-line treatment [11]. The advantage of our study compared to previously published studies is the time-dependent evaluation of sFLC based on serial sFLC measurements.

Interestingly, Abdallah et al. were able to demonstrate that in MM patients who achieve CR and an absence of clonal bone marrow plasma cells in the multiparametric flow cytometry, a pathologic FLCr due to a suppression of the involved, the uninvolved or both sFLC is accompanied with the same outcome compared to a normal FLCr [35]. In contrast, a pathologic FLCr due to an increase in the involved sFLC is associated with a worse outcome. Unfortunately, in our analysis, the reason for an abnormal FLCr was not investigated. However, due to the fact that a pathologic FLCr not related to an increase in the involved sFLC would be classified as “FLCr normalization”, a further strengthening of our results would be expected.

In our analysis, the positive prognostic effect of time-dependent FLCr normalization occurred irrespective of whether the achievement of FLCr normalization was lost in between. This is in contrast to the worse prognostic impact of a loss of CR or minimal residual disease (MRD). However, this might be explained by the often therapy-induced abnormal FLCr. This is supported by the work of Abdallah et al., who demonstrated that in more than half of the patients with an abnormal FLCr, this is caused by a suppression of sFLC [35].

Surprisingly, the response after consolidation (CR vs. non-CR) did not show a significant effect on PFS and OS in our time-dependent multivariable Cox regression model. This is in contrast to previous studies demonstrating an association between CR and a superior outcome [36,37]. An explanation for this could be that the achievement of a time-dependent FLCr normalization is a better predictor for survival than a response at a single time point. However, it has to be noticed that the rate of CR after consolidation in the MM5 trial is underestimated because bone marrow punctures were not obligatory. Besides the 28.3% of the patients reaching CR after consolidation, there was a further 28.5% reaching near CR (nCR). An impact of the underestimation of CR and a low CR rate after consolidation (126/590) cannot be excluded. Furthermore, it can be assumed that due to missing maintenance therapy in patients with CR after consolidation in arms B, the positive prognostic impact of CR was weakened.

A relation between sFLC and the IMWG response can be explained by the disease itself. A progression of the disease goes along with an increase in the monoclonal protein but also the involved sFLC, affecting FLCr. Furthermore, the percentage of patients with a normal FLCr increases with the improving response category [11]. However, independent of the response, the positive prognostic impact of FLCr normalization during MM therapy remains. 

Because of the effective novel agent-based therapies, there is a need for more precise techniques to detect a residual disease that could be missed by the determination of conventional remission alone. In this context, the determination of MRD by multiparametric flow cytometry or next-generation sequencing is of increasing importance [9], and the results on MRD within the MM5 trial will be presented separately.

After an effective tumor load reduction and the completion of intensive treatment, the recovery of bone marrow and immune system functions leads to a physiological production of polyclonal sFLC and immunoglobulins by plasma cells. In the current analysis, immune reconstitution occurred later than FLCr normalization. This is likely due to the intensive treatment and the strict need for the normalization of all uninvolved immunoglobulins to fulfil this criterion. Immune reconstitution is therefore achieved by a smaller percentage of patients with a maximum of 32.2% after consolidation. González-Calle et al. showed similar rates six months after ASCT, further increasing to 52% one year after ASCT [16]. A possible explanation could be the difference in maintenance strategies: Compared to lenalidomide maintenance therapy for two years (arms A1 and A2) or until the achievement of CR (arms B1 and B2) in the MM5 trial, in the Spanish study, only 57% of the patients received maintenance therapy, which consisted of 80% of the cases of Interferon-α [16]. Furthermore, Jimenez-Zepeda et al. were able to demonstrate that patients receiving a lenalidomide-based consolidation therapy have a lower rate of immune reconstitution one year after ASCT [38].

We demonstrate that a time-dependent immune reconstitution before the start of maintenance is significantly associated with prolonged PFS and OS. Furthermore, there is a strong effect of immune reconstitution after consolidation on OS, representing the process of physiological B-cell reconstitution after ASCT. This prognostic effect of immune reconstitution can be mainly explained by the fact that immune reconstitution is a marker of the treatment response. During the course of the disease, progressive disease is much more frequent than non-relapse mortality [39]. The most common cause of death due to progressive disease is an infection caused by immunoparesis [40].

Similarly, González-Calle et al. were able to demonstrate a positive prognostic impact by achieving immune reconstitution. However, this impact was seen in the landmark analysis one year after ASCT and not in the landmark analyses conducted at earlier time points [16]. These results are in line with two other studies and can be explained by completed B-cell reconstitution one year after ASCT [38,41,42]. It has to be noted that, currently, there are only a few retrospective studies on the prognostic effect of immune reconstitution after ASCT and that further studies are needed to clarify the optimal time point of assessment.

## 5. Conclusions

To conclude, two-thirds of the patients with newly diagnosed secretory MM treated within the MM5 trial achieved normalization of the FLCr after consolidation therapy. A time-dependent FLCr normalization at any time point prior to the start of maintenance therapy significantly prolonged PFS and OS independent of age, ISS, cytogenetics, treatment arm and the response after consolidation. Furthermore, a time-dependent immune reconstitution during therapy predicted superior PFS and OS. These results suggest that FLCr normalization and immune reconstitution during therapy constitute important and simple to assess prognostic factors for patients with MM. Therefore, we recommend the serial assessment of sFLC and immunoglobulins during MM therapy.

## Figures and Tables

**Figure 1 cancers-13-04856-f001:**
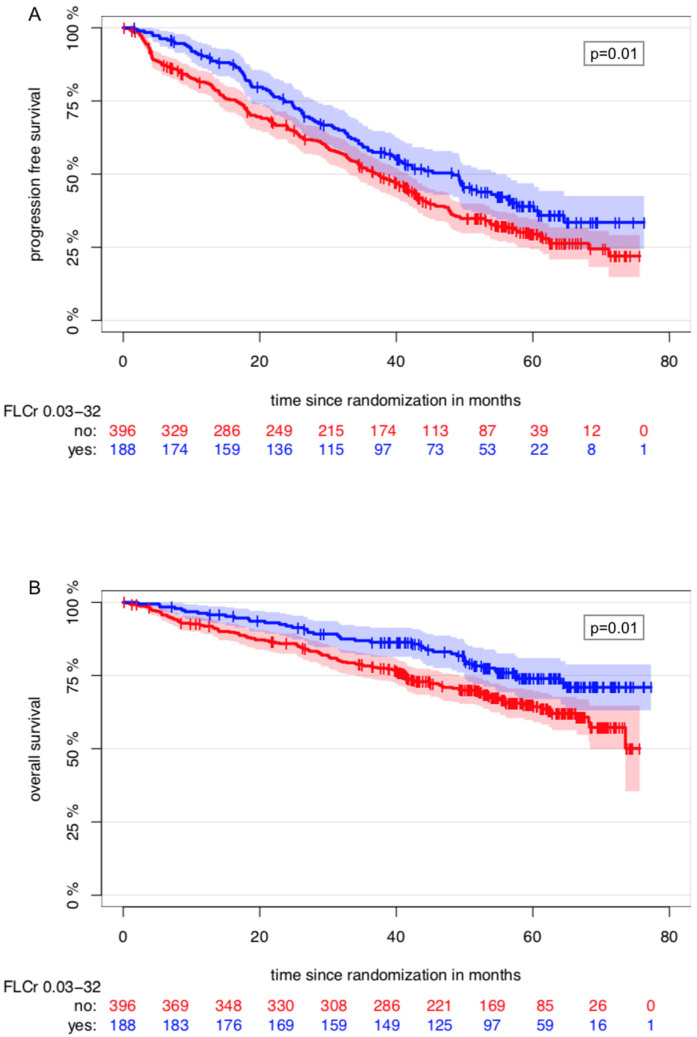
Kaplan–Meier Estimator of the impact of the FLCr at baseline on PFS (**A**) and OS (**B**).

**Figure 2 cancers-13-04856-f002:**
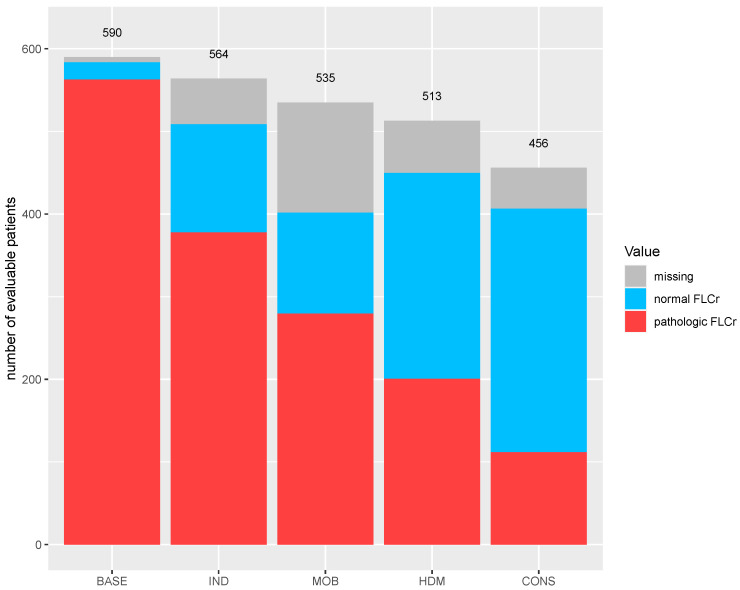
FLCr normalization during first-line therapy in the MM5 trial. The number of evaluable patients is represented by the number of patients on study at the start of each treatment phase. Abbreviations: BASE, baseline; CONS, consolidation; HDM, high-dose therapy with melphalan; IND, induction; MOB, mobilization.

**Figure 3 cancers-13-04856-f003:**
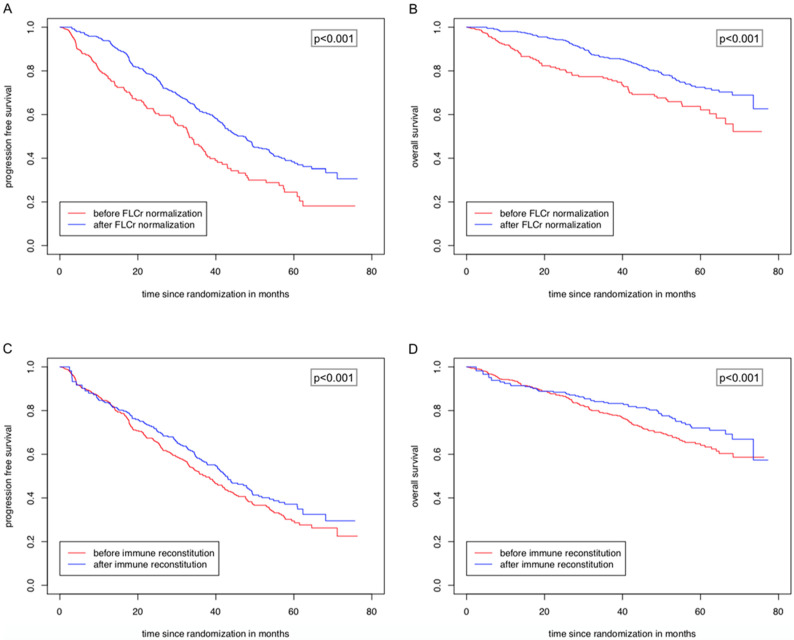
Simon–Makuch Plots showing the risk of progression (**A**,**C**) and death (**B**,**D**) depending on the achievement of FLCr normalization and immune reconstitution. The survival times until a potential achievement of FLCr normalization/immune reconstitution and after a potential achievement are separately shown in two curves. Therefore, one patient can be found in two curves. For example, patients with immunoparesis who achieve immune reconstitution during therapy start in the curve “before immune reconstitution” and switch after the achievement of immune reconstitution in the curve “after immune reconstitution”.

**Figure 4 cancers-13-04856-f004:**
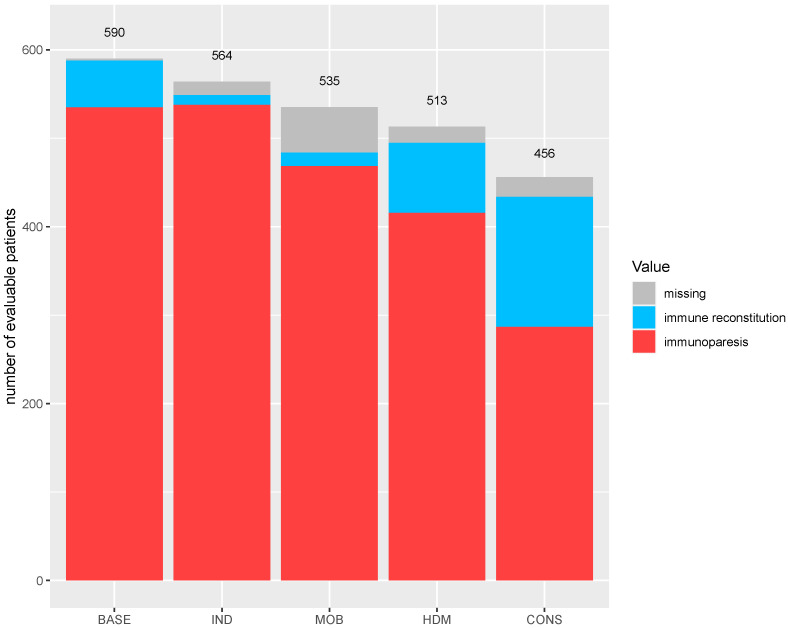
Immune reconstitution during first-line therapy in the MM5 trial. The number of evaluable patients is represented by the number of patients on study at the start of each treatment phase.

**Table 1 cancers-13-04856-t001:** Baseline characteristics of the analysis population.

Variable	*n* (*n* = 590)	%
Sex (*n* = 590)		
Female	242	41.0
Male	348	59.0
Age (*n* = 590)		
Median (range)	59 (32–70) years	
Myeloma subtype (*n* = 590)		
IgG	364	61.7
IgA	121	20.5
Bence-Jones	105	17.8
Light chain isotype (*n* = 590)		
Kappa	402	68.1
Lambda	188	31.9
Calcium (*n* = 590)		
>2.65 mmol/L	79	13.4
≤2.65 mmol/L	511	86.6
Renal insufficiency * (*n* = 590)		
Yes	66	11.2
No	524	88.8
Hemoglobin (*n* = 590)		
<10 g/dL	304	51.5
≥10 g/dL	286	48.5
Bone disease ** (*n* = 590)		
Yes	534	90.5
No	56	9.5
ISS (*n* = 590)		
I	226	38.3
II	204	34.6
III	160	27.1
Adverse cytogenetics *** (*n* = 527)		
Yes	267	50.7
No	260	49.3
LDH (*n* = 588)		
<308 U/L	551	93.7
≥308 U/L	37	6.3
Abnormal FLCr (*n* = 584)		
Yes	563	96.4
No	21	3.6
Immunoparesis (*n* = 588)		
Yes	535	91.0
No	53	9.0

Abbreviations: ISS, International Staging System; LDH, lactat dehydrogenase; FLCr, free light chain ratio. * Creatinine > 2 mg/dL and/or glomerular filtration rate < 40 mL/min. ** One or more osteolytic lesions. *** Deletion 17p13, translocation t(4;14) or gain 1q21 more than three copies.

**Table 2 cancers-13-04856-t002:** Impact of achieved FLCr normalization until the start of maintenance at the latest on PFS and OS. The results of the time-dependent multivariate Cox regression analysis.

Variable	PFS		OS	
	HR (95% CI)	*p*-Value	HR (95% CI)	*p*-Value
Age (per year)	1.00 (0.98–1.01)	0.89	1.02 (1.00–1.04)	0.06
ISS (II vs. I)	1.46 (1.13–1.90)	<0.01	1.77 (1.16–2.71)	0.01
ISS (III vs. I)	1.82 (1.37–2.43)	<0.01	2.91 (1.90–4.45)	<0.01
Adverse cytogenetics (yes vs. no)	2.12 (1.68–2.68)	<0.01	2.96 (2.07–4.23)	<0.01
Treatment arm (B vs. A)	1.04 (0.82–1.31)	0.75	1.56 (1.12–2.17)	0.01
Response after CONS (CR vs. non-CR)	1.04 (0.76–1.40)	0.82	0.71 (0.44–1.15)	0.16
FLCr normalization (yes vs. no)	0.61 (0.47–0.79)	<0.01	0.67 (0.48–0.93)	0.02

Abbreviations: CR, complete remission; HR, hazard ratio; OS, overall survival; PFS, progression-free survival; 95% CI, 95% confidence interval.

**Table 3 cancers-13-04856-t003:** The impact of achieved immune reconstitution until the start of maintenance at the latest on PFS and OS. The results of the time-dependent multivariate Cox regression analysis.

Variable	PFS		OS	
	HR (95% CI)	*p*-Value	HR (95% CI)	*p*-Value
Age (per year)	1.00 (0.98–1.01)	0.73	1.02 (1.00–1.04)	0.05
ISS (II vs. I)	1.43 (1.10–1.85)	0.01	1.72 (1.13–2.63)	0.01
ISS (III vs. I)	1.88 (1.41–2.50)	<0.01	2.97 (1.94–4.53)	<0.01
Adverse cytogenetics (yes vs. no)	2.09 (1.66–2.63)	<0.01	2.97 (2.09–4.24)	<0.01
Treatment arm (B vs. A)	1.04 (0.83–1.30)	0.75	1.55 (1.11–2.16)	0.01
Response after CONS (CR vs. non-CR)	0.98 (0.73–1.30)	0.87	0.65 (0.41–1.04)	0.07
Immune reconstitution (yes vs. no)	0.77 (0.60–0.99)	0.04	0.59 (0.41–0.86)	0.01

## Data Availability

The data presented in this study are available upon request from the corresponding author. The data are not publicly available due to privacy issues.

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
