# Peer review of "Prognostic Impact of Serum Free Light Chain Ratio Normalization in Patients with Multiple Myeloma Treated within the GMMG-MM5 Trial"

_cancers, 2021, doi:10.3390/cancers13194856_

Round 1
Reviewer 1 Report
The findings presented in this manuscript are informative for physicians who are involved in daily practical works. This paper proposes “serum FLC normalization” as a novel categorical response in multiple myeloma regardless of stringent CR, where bone marrow biopsy is required.
However, the data showing the relationship between “serum FLC normalization” and IMWG responses (i.e., PR, CR, and sCR) is helpful for readers to understand the findings in this paper.
Reviewer 2 Report
The manuscript is well written, and the data are interesting and important. There is some extra information I would like to see, and some additional topics for discussion. See below
Line 110. It is probably not the excessive production of monoclonal protein that leads to immunoparesis, but rather the excess of clonal plasma cells in the bone marrow, displaces normal plasma cells.
Table 1: It’s a bit strange that “Bence-Jones” is figuring as a “Heavy Chain isotype”. Maybe rephrase the sub-heading?
I miss information of the renal failure cutoff in the inclusion criteria. Could be included in the legend of table 1.
Why is the adverse cytogenetics different from the standard in R-ISS? Especially, why is 14;16 missing.
Line 219++. It is shown that arms B had shorter OS than arms A. I would like to know in which arms the treatment was longer. This is an important finding by itself, even though on the side of the main topic in this manuscript.
Figure 3: I don’t understand the phrases “before” and “after” normalization/reconstitution. Isn’t the comparison between patients who did or did not achieve this?
In M&M, it is mentioned that FLC was also evaluated throughout the maintenance treatment. These data are not part of the Results section? Why not? It would be interesting to see if also in this period, more patients are achieving FLCr normalization.
Line 290: Could add that when you define certain time points of FLC normalization, the groups achieving this are maybe too small to be statistically significant, and mention that this contrasts with the Lopez-Anglada paper which was mentioned in the introduction.
Line 290++. I think the authors should propose continued FLC measurements also after normalization, in line with their following comment of light chain escape. Especially when the IMWG recommendation with continuous urine electrophoresis is not followed, which most doctors don’t do in routine practice.
299: The “first best response” is not a very good phrase. The reason is that this is something you can only know in retrospect. In real-time you never know when the “best response” is achieved. Can you find another way of phrasing this?
To be discussed:
There is a paper showing that abnormal FLCr if both light chain types are normal/low, has the same prognostic impact as normal FLCr. This should be mentioned and discussed.
Since CRs (and hence sCRs) in this study is underestimated, could it be that normalization of FLCr is associated to CR/sCR, and hence is not an independent marker? Should be discussed. And the same goes for association with MRDnegativity, which is not presented here.
It should be discussed what the link is between immune reconstitution and overall survival. Is it just a marker of tumor load/efficacious treatment? Or is it because the non-reconstituted are more prone to infections? Or less tolerant to treatment because of partial bone marrow failure? Should this lead to more aggressive IVIG treatment? Are we using this enough?
In the conclusion the authors mention both the FLCr results and the immunoglobulin results. But the conclusion ends with only recommending FLC measurements. Why not include immunoglobulins in this sentence.
Ortography:
It is more common to write FLCr than rFLC
It definitely is more common to write ASCT instead of ABSCT.
Line 56: Remove the word «one».
Line 59: Add the word “way” or something like that, after the word “simple”.
Reviewer 3 Report
Klein et al. present an interesting article that highlights the importance of sequential rFLC measurement and shows that achievement of rFLC normalization at any time point is a favorable prognostic factor. The manuscript is very well written and the analysis is well done.
Some questions/concerns remain-
- The authors claim that a normalization or the rFLC is a good prognostic sign, even if normalization is lost afterwards. That seems a bit contradictory to other markers of response, such as CR or MRD negativity, where loss of these leads to poorer outcome. How do the authors explain that patients whose rFLC becomes abnormal still have good outcome? Is the loss of a normal rFLC the results of a decreased uninvolved light chain?
- The number of patients that entered the trial is 590, yet only 456 are available for evaluation after consolidation. What happened to the remaining 134 patients? Have they been lost to follow up?
- The fact that CR is not prognostic in the multivariate model is probably also due to part of patients who achieved CR after consolidation not proceeding with maintenance (as per study protocol). This should be discussed.
Minor issues
- Please add p values to Figures 1 and 3
- For Figures 2 and 4, please explain yes (blue color) and no (red color) in the legend or substitute with rFLC normalization or non-normalization.
Reviewer 4 Report
The authors report the impact of serum rFLC normalization in patients with multiple myeloma; normalization of serum rFLC at any one time before start of maintenance is significantly associated with prolonged PFS and OS.
1) What is the main question addressed by the research?
R: Normalization of rFLC is a good prognostic parameter.
2) Do you consider the topic original or relevant in the field, and if so, why?
R: The topic is relevant in the field (perhaps less original).
3-4)Are the conclusions consistent with the evidence and arguments presented and do they address the main question posed?
YES Are the references appropriate? YES
5) Please include any additional comments on the tables and figures.
R: No comments on tables; about figures 2 and 4, legend should be clearer.
